# Diagnostic Value of FDG PET/CT in Surveillance after Curative Resection of Breast Cancer

**DOI:** 10.3390/cancers15092646

**Published:** 2023-05-07

**Authors:** Hwanhee Lee, Joon Young Choi, Yeon Hee Park, Jeong Eon Lee, Seok Won Kim, Seok Jin Nam, Young Seok Cho

**Affiliations:** 1Samsung Medical Center, Department of Nuclear Medicine, Sungkyunkwan University School of Medicine, Seoul 06351, Republic of Korea; hh2020.lee@samsung.com (H.L.); jynm.choi@samsung.com (J.Y.C.); 2Samsung Medical Center, Division of Hematology-Oncology, Department of Medicine, Sungkyunkwan University School of Medicine, Seoul 06351, Republic of Korea; yeonh.park@samsung.com; 3Samsung Medical Center, Department of Surgery, Sungkyunkwan University School of Medicine, Seoul 06351, Republic of Korea; jeongeon.lee@samsung.com (J.E.L.); seokwon1.kim@samsung.com (S.W.K.); seokjin.nam@samsung.com (S.J.N.)

**Keywords:** breast cancer, surveillance, FDG PET, routine follow-up, diagnostic value

## Abstract

**Simple Summary:**

With increasing incidence of breast cancer and improvement in treatment, the concern about surveillance management after curative treatment in patients with breast cancer also has increased. However, there is a debate regarding routine surveillance follow-up examinations in clinical practice. This retrospective study was designed to evaluate the diagnostic value of routine surveillance FDG PET/CT in patients with breast cancer. A total of 2121 scans in 1681 women with breast cancer in surveillance were included in the study. The frequency of positive surveillance PET/CT findings was 5.0% (105 of 2121 scans) and the sensitivity, specificity, positive predictive value, negative predictive value, and accuracy for detecting clinically unsuspected recurrent breast cancer or second primary malignancy were 100%, 98.5%, 70.5%, 100% and 98.5%, respectively. In conclusion, surveillance fluorodeoxyglucose PET/CT showed superior diagnostic performance and comparable rate of positive findings with conventional imaging modality in the detection of clinically unexpected recurrent breast cancer after curative treatment.

**Abstract:**

With increasing incidence of breast cancer and improvement in treatment, the concern about surveillance management also has increased. This retrospective study was designed to evaluate the diagnostic value of routine surveillance FDG PET/CT in patients with breast cancer. The diagnostic performance of surveillance PET/CT was analyzed regarding sensitivity, specificity, positive predictive value, negative predictive value and accuracy. The diagnostic accuracy was defined as the ability to differentiate recurrence and no-disease correctly and the proportion of true results, either true positive or true negative, in the population. Findings from pathologic examination; other imaging modalities such as CT, MRI and bone scan; or clinical follow-up were used as the reference standard. In this study of 1681 consecutive patients with breast cancer who underwent curative surgery, surveillance fluorodeoxyglucose PET/CT showed good diagnostic performance in the detection of clinically unexpected recurrent breast cancer or other malignancy, with a sensitivity of 100%, specificity of 98.5%, positive predictive value of 70.5%, negative predictive value of 100% and accuracy of 98.5%. In conclusion, surveillance fluorodeoxyglucose PET/CT showed good diagnostic performance in the detection of clinically unexpected recurrent breast cancer after curative surgery.

## 1. Introduction

Breast cancer is one of the most common malignancies newly diagnosed in women worldwide. However, thanks to advances in early diagnosis and treatment, the associated mortality has decreased. The 10-year survival rate of patients with breast cancer with local disease has been recorded at 70–80%, and even as high as 90% [1,2]. Along with these facts, the concerns about surveillance management after primary curative treatment of breast cancer have increased. Although it is not universally recommended, in current clinical practice, the use of FDG PET/CT in routine surveillance follow-up has been increasing to detect asymptomatic breast cancer recurrence that is potentially amenable to treatment.

It is well-known that fluorine-18 fluorodeoxyglucose (FDG) PET/CT is the most valuable imaging modality in detecting recurrence or metastasis in patients with breast cancer with documented or suspected recurrence detected at physical, laboratory, or other conventional imaging examinations [3,4,5]. With excellent diagnostic sensitivity and accuracy, FDG PET/CT can provide clarification on whether the recurrence is isolated, which potentially alters treatment options. On the other hand, current guidelines recommend against routine FDG PET/CT imaging for breast cancer surveillance of asymptomatic patients because of the high cost and lack of evidence. To our knowledge, with the exception of a study with a small number of subjects [4], no previous studies investigated the diagnostic value of routine surveillance FDG PET/CT primarily for detecting recurrent breast cancer. In the Republic of Korea, between 2006 and 2016, surveillance FDG PET/CT in cancer patients was covered by National Healthcare Insurance after curative therapy even without clinical suspicion of recurrence [6].

Therefore, with a large population, we performed this study to evaluate the diagnostic value of routine surveillance FDG PET/CT in the detection of clinically unsuspected recurrent breast cancer after primary curative treatment.

## 2. Materials and Methods

### 2.1. Study Cohort

Our institutional review board approved this retrospective study, and the requirement for informed consent was waived (2021-06-006). We reviewed medical records from 2742 consecutive patients diagnosed with breast cancer who underwent FDG PET/CT between 2006 and 2016 at our institution. Seven patients were excluded because the surgical histopathological finding was not accessible. Men (*n* = 24), patients with ductal carcinoma in situ and breast cancer other than invasive carcinoma (*n* = 383) and patients with stage IV disease with distant metastasis (*n* = 647) were also excluded. We screened all PET/CT scans of the included 1681 patients to identify the indication or reason for scanning at the time of imaging. In this study, “surveillance FDG PET/CT” refers to a routine follow-up scanning after primary curative surgery without documentation or suspicion of recurrence at conventional imaging, laboratory tests and clinical symptoms and signs. Among 2197 surveillance PET/CT scans, 74 scans obtained within 3 months after surgery (the time between scanning and surgery was too short to avoid hypermetabolic inflammation) and 2 scans of patients whose follow-up durations were shorter than 12 months (predefined minimally required follow-up period to decide whether a negative finding was true negative or not) after imaging were excluded. The flow diagram of inclusion and exclusion criteria is presented in Figure 1.

### 2.2. Medical Report Review

In our institution, we recommend that breast cancer survivors visit the clinic annually for 5 to 10 years after curative tumor resection. We reviewed medical records from the included patients, including clinical characteristics, initial disease stage, mode of tumor resection, histopathologic finding, protocol of curative treatment, disease response to neoadjuvant chemotherapy (NAC) and onset and diagnosis of recurrence. Disease stage of breast cancer was based on the clinical prognostic stage suggested by the American Joint Committee on Cancer Staging Manual, 8th edition [7], including the FDG PET/CT finding. The tumor responses to NAC were categorized into three categories according to the definition of the recent National Comprehensive Cancer Network guidelines [8].

### 2.3. Image Acquisition and Analyses

Patients fasted for at least 6 h prior to FDG PET/CT, and the blood glucose level was restricted to <200 mg/dL at the time of FDG injection. Imaging was performed after 60 min of FDG distribution (5 MBq/kg) using a Discovery LS (GE Healthcare) or a Discovery STe (GE Healthcare) PET/CT scanner. CT scanning was performed with a continuous spiral technique and followed by PET scanning from base of skull to mid-thigh. 

The PET/CT images were interpreted by two nuclear medicine physicians (Lee, H. and Choi, J.Y., with, respectively, > 6 years and > 20 years of experience in FDG PET reading). Imaging interpretation was based on visual inspection with semiquantitative analyses and comparisons with initial or prior PET/CT studies. Sites of abnormal metabolic activity were classified as either suspicious for malignancy or benign by applying the following image interpretation criteria: (a) Focal abnormally increased FDG uptake that exceeded that in the surrounding tissue and was discernible from physiologic uptake or benign disease was principally considered as possible malignancy. (b) In case of lymph nodes, hypermetabolic nodes with calcification or high attenuation in CT images were interpreted as benign nodes [9]. (c) For pulmonary metastasis, newly developed nodules or those that increased in size in CT images with discernible FDG uptake from surrounding parenchyma were interpreted as metastasis.

### 2.4. Diagnostic Performance Analyses

To evaluate diagnostic value of surveillance FDG PET/CT by means of sensitivity, specificity, positive predictive value (PPV), negative predictive value (NPV) and accuracy, the imaging findings were tagged as true positive, false positive, true negative, or false negative. The final decision as to whether the positive findings were a true recurrence or not was mainly based on pathologic examination of biopsy specimens. In the case of positive scans without relevant biopsy procedure, clinical impression with further image study and other clinical correlation were reviewed. Clinical follow-up for 12 months after each PET/CT examination was reviewed to confirm negative findings as true or false. The frequency of positive findings and diagnostic performance were compared according to clinical variables including time between imaging and surgery (2 <, 2–4 and 4 < years), initial clinical prognostic stage (stage I, II and III), option of NAC (surgery only and NAC before surgery) and response to neoadjuvant chemotherapy (no, partial and complete response). The impact of surveillance PET/CT findings on patients’ management was also analyzed.

### 2.5. Statistical Analysis

The chi-squared test or Fisher’s exact test (employed when sample sizes were small) were used for comparison of diagnostic performance according to clinical variables. Data were analyzed by MedCalc, ver.15.8 (MedCalc Software bvba., Ostend, Belgium), and two-sided *p*-values < 0.001 were accepted as a significant difference [10].

## 3. Results

### 3.1. Characteristics of Study Population

A total of 2121 surveillance FDG PET/CT scans in 1681 women (mean age of 48 years ± 9 at diagnosis) with breast cancer were included in the study. Demographics and clinical characteristics of included patients are shown in Table 1.

### 3.2. Diagnostic Findings and Results of Surveillance FDG PET/CT

Among the 2121 surveillance FDG PET/CT scans, 105 scans (5.0%) showed positive findings suggesting malignant lesion(s). Of the 105 positive scans, 74 scans (70.5%) were confirmed as true positive and 31 as false positive (29.5%). True-positive findings included detection of second primary malignancy (*n* = 6) as well as recurrent breast cancer. The most frequent type of breast cancer recurrence correctly detected was distant metastasis (*n* = 50). Figure 2 shows examples of local recurrence, distant metastasis of recurrent breast cancer and second primary thyroid cancer correctly detected with surveillance PET/CT.

In deciding on true recurrence, hypermetabolic lesions in 46 scans were pathologically confirmed by fine needle aspiration or tissue biopsy, and lesions in 28 scans were finally determined to be recurrences by confirmation through another radiology imaging modality or follow-up PET/CT. Of the 31 scans that showed false-positive findings, lesions in 25 scans were confirmed as benign by biopsy, lesions in 4 positive scans were clinically confirmed as benign by further evaluation with other image study (CT or MRI) and bone lesions in 2 positive scans were considered as benign with patients’ relevant trauma history and were proved to not be recurrences in 12-month follow-up. Regional lymph node was the most erroneously detected lesion by surveillance PET/CT. Ipsilateral axillary lymph nodes with increased FDG uptake in 11 scans turned out to be benign reactive or inflammation by biopsy. Table 2 shows details of positive findings of surveillance PET/CT. There was no false-negative scan pathologically or clinically proved. 

Therefore, the sensitivity, specificity, PPV, NPV and accuracy of surveillance FDG PET/CT for detecting clinically unsuspected recurrent breast cancer or second primary malignancy were 100% (74 of 74 scans), 98.5% (2016 of 2047 scans), 70.5% (74 of 105 scans), 100% (2016 of 2016 scans) and 98.5% (2090 of 2121 scans), respectively. The diagnostic accuracy is the proportion of true results, either true positive or true negative, in the population; (true positive + true negative) / (true positive + true negative + false positive + false negative).

### 3.3. Diagnostic Performance of Surveillance FDG PET/CT According to Several Clinical Variables

Table 3 shows the frequency of positive findings and the diagnostic efficacy of FDG PET/CT according to the time between curative surgery and imaging, initial clinical prognostic stage, option of NAC and disease response to NAC. Since no false-negative PET/CT findings were proven with pathologic or clinical examination, the sensitivities and NPVs of the following subgroups were all 100%. Subsequent diagnostic performance was analyzed based on scan (raw data in parentheses are the number of scans).

The frequency of positive findings of surveillance PET/CT at different times between curative surgery and scanning was 6.0% (44 of 729) for two or fewer years, 4.3% (35 of 818) for 2–4 years and 4.4% (25 of 574) for four or more years, which showed no difference (*p* = 0.11). There was also no difference between the three groups in specificity, PPV and accuracy: 98.0% (685 of 699) vs. 99.0% (782 of 790) vs. 98.4% (549 of 558) for specificity (*p* = 0.16); 68% (30 of 44) vs. 78% (28 of 36) vs. 64% (16 of 25) for PPV (*p* = 0.34); 98.1% (715 of 729) vs. 99.0% (810 of 818) vs. 98.4% (565 of 574) for accuracy (*p* = 0.20). The sensitivity and NPV results were all 100%.

The frequency of positive PET/CT findings in stage III (8.6%, 63 of 732) was higher than that in stages I (2.8%, 25 of 903) and II (3.5%, 17 of 486) (*p* < 0.001). However, the diagnostic performance was not different between stage I, II and III: 98.9% (878 of 888) vs. 99.4% (469 of 472) vs. 97.5% (669 of 687) for specificity (*p* = 0.10); 60% (15 of 25) vs. 82% (14 of 17) vs. 71% (45 of 63) for PPV (*p* = 0.39); 98.9% (893 of 903) vs. 99.4% (483 of 486) vs. 97.5% (714 of 732) for accuracy (*p* = 0.11). The sensitivity and NPV results were all 100%.

The frequency of positive PET/CT findings in patients who underwent NAC before surgery (9.0%, 31 of 345) was higher than that in patients undergoing surgery without NAC (4.2%, 74 of 1776) (*p* < 0.001). However, the diagnostic performance was not different between the two groups (NAC + surgery vs. surgery only): 96.9% (314 of 324) vs. 98.8% (1702 of 1723) for specificity (*p* = 0.01); 68% (21 of 31) vs. 72% (53 of 74) for PPV (*p* = 0.87); 97.1% (335 of 345) vs. 98.8% (1755 of 1776) for accuracy (*p* = 0.03). The sensitivity and NPV results were all 100%.

Among 51 scans of patients who achieved complete response after NAC, only 1 scan revealed a positive finding, which was proven to be a true positive; the rest were all true-negative scans, with 100% of the specificity, PPV and accuracy. The specificity, PPV and accuracy of scans in patients who failed to achieve complete response after NAC were 96.4% (264 of 274, *p* = 0.36), 67% (20 of 30, *p* = 0.70) and 96.6% (284 of 294, *p* = 0.38), respectively.

### 3.4. Impact of Surveillance FDG PET/CT on Patient Management

Of a total of 2121 included surveillance PET/CT scans, 77 scans (3.6%) led to a change in management of the patient; 1.8% (16 of 903 scans) in stage I, 3.1% (15 of 486 scans) in stage II and 6.2% (46 of 732 scans) in stage III disease. Among the 105 positive scans, all true-positive and three false-positive findings at PET/CT led to a change in patients’ intended management (73.3%, 77 of 105 scans). True-positive surveillance PET/CT was followed by a curative surgery, salvage chemotherapy, or radiotherapy. Three patients underwent surgical resection of hypermetabolic tissue classified as recurrence at surveillance PET/CT despite knowing that the positive findings were false by prior biopsy. The second primary malignancies depicted with surveillance PET/CT were all in early stages and curatively treated.

## 4. Discussion

With increasing incidence of breast cancer and improvement in treatment, the concern on surveillance management has increased. However, there is a debate regarding routine surveillance follow-up examination. Our study showed the clinical value of surveillance FDG PET/CT in detecting early disease recurrence or unexpected second malignancies in patients with breast cancer with high diagnostic efficacy irrespective of clinical variables. Although overall frequency of positive PET/CT findings was not high (5.0%, 105 of 2121 scans), the sensitivity, specificity, PPV, NPV and accuracy were excellent: 100% (68 of 68 scans), 98.5% (2016 of 2047 scans), 69% (68 of 99 scans), 100% (2016 of 2016 scans) and 98.5% (2084 of 2115 scans), respectively. Moreover, surveillance PET/CT correctly depicted unsuspected distant metastases and second primary malignancy not detectable with other conventional imaging modalities, as well as asymptomatic local–regional recurrences with curable potential. 

For surveillance imaging practice in breast cancer, the current guidelines recommend against routine FDG PET/CT [1,2]. In these circumstances, previous studies on surveillance PET/CT in breast cancer were limited to evaluating diagnostic performance of PET/CT optionally performed with suspicion and comparing diagnostic performance of PET/CT versus that of conventional imaging modalities [3,5]. To the best of our knowledge, this study is the first to investigate the value of routine surveillance FDG PET/CT of curatively treated breast cancer in a large cohort. In a research work with subsequent surveillance PET/CT (regardless of prior evidence of recurrence) in breast cancer, the overall respective sensitivity, specificity, PPV, NPV and accuracy were 97.7%, 98.1%, 98.8%, 96.3% and 97.9%. Of the included fourth and subsequent scans, 134 scans were obtained without a clinical suspicion of recurrence. The diagnostic sensitivity, specificity, PPV, NPV and accuracy of the 134 scans were 93.3%, 99.2%, 93.3%, 99.2% and 98.5%, respectively, which showed comparable diagnostic performance to our study except in PPV (93.3% vs. 69% in our study). They categorized indeterminate findings (FDG-avid lesions, but not clearly identified as recurrence) as ‘negative’, which may lead to higher PPV values than those in our study [4]. In our study, the sensitivity and NPV were 100%, which means that negative findings of surveillance PET/CT might be also clinically valuable by reassuring that a patient did not have recurrence or another malignancy.

Mammography is the only surveillance imaging modality universally recommended for breast cancer, although US is another popularly used routine practice for surveillance. Studies have reported that the rate of recurrence detected only with surveillance mammography was as low as 8% [11], with unreliable sensitivity (51.1%) [12]. In addition, the rate of positive findings of surveillance US was 2.9% (57 of 1968 and 85 of 2925 cases) in two studies [13,14], with a sensitivity of 45.9–95.8%. A study reported higher diagnostic performance of MRI in breast cancer surveillance than that of mammography and US, but the rate of positive MRI was 5.4% (110 of 2026) [15], which is similar with that of PET/CT shown in our study. Therefore, the diagnostic performance of surveillance PET/CT in our study (100% for diagnostic sensitivity and 98.5% for accuracy) was superior to that of conventional imaging modalities in previous studies, with comparable rates of positive findings. Moreover, PET/CT demonstrates distant metastasis in a single examination with efficacy on restaging of recurrent breast cancer, whereas conventional images are limited to the detection of local–regional recurrence [16]. All second primary malignancies correctly depicted at surveillance PET/CT in our study were in a curable, early stage, which is similar to previous studies in which cure may be expected after therapy [17].

Our study has several limitations. First, this study is retrospective in design, which result in relevance bias. In surveillance management for patients with breast cancer, the routine surveillance laboratory and imaging tests were performed at the clinician’s discretion or patient’s preference rather than by a routine protocol. However, during the study period, most patients underwent one or more surveillance PET/CT during surveillance with coverage of National Healthcare Insurance. Second, not all lesions were evaluated to confirm recurrent disease if multiple lesions were detected by a scan. Third, two different kinds of PET/CT scanners were used in our study. However, in a previous study using the same kinds of scanners for esophageal cancer, it was reported that the sensitivity and specificity of the two scanners were not significantly different (*p* = 0.087) [18]. Therefore, a prospective study, such as a randomized controlled trial that examines the cost-effectiveness of routine surveillance PET/CT, is warranted.

## 5. Conclusions

In conclusion, surveillance FDG PET/CT showed an excellent diagnostic performance and a comparable rate of positive findings with conventional imaging modality in the detection of clinically unexpected recurrent breast cancer after curative resection. A well-designed prospective study on survival benefit and cost-effectiveness of routine surveillance PET/CT in breast cancer is warranted.

## Figures and Tables

**Figure 1 cancers-15-02646-f001:**
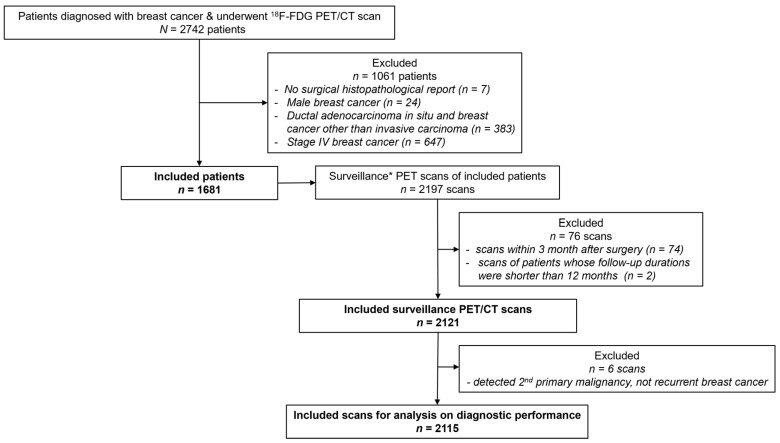
Flow diagram of the inclusion and exclusion criteria. Surveillance* PET/CT is defined as a routine follow-up scanning after primary curative surgery without documentation or suspicion of recurrence at conventional imaging, laboratory tests, and clinical symptoms and signs.

**Figure 2 cancers-15-02646-f002:**
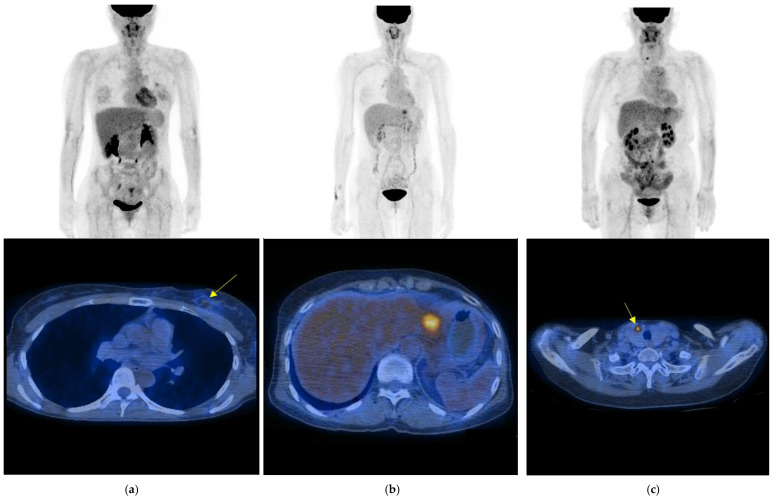
Maximum intensity projection image (top) and axial fusion image of PET/CT (bottom) of unexpected malignancies depicted with surveillance PET/CT. (**a**) Images of 38-year-old woman patient with breast cancer show a focal FDG uptake in upper inner quadrant of left breast (arrow) confirmed as local recurrence (obtained 12 months after curative surgery). (**b**) Images of 62-year-old woman patient with breast cancer show distant metastasis to liver (obtained 31 months after curative surgery). (**c**) Images of 56-year-old woman patient with breast cancer show unexpected second primary thyroid cancer (arrow) proven as papillary thyroid cancer stage I (obtained 65 months after curative surgery).

**Table 1 cancers-15-02646-t001:** Demographics and clinical characteristics of patients.

Characteristics	No. of Patients
Included female patients with breast cancer	1681
Age at diagnosis (year)	
<40	309
40~50	765
51~60	454
>60	153
Mean age (SD): 48 (9)	
Primary tumor location	
Right breast	642
Left breast	552
Bilateral breast	487
Clinical prognostic stage (AJCC 8th ed.)	
Stage I	767
Stage II	397
Stage III	517
Protocol of curative treatment	
NAC + Surgery	4
NAC + Surgery + AC	16
NAC + Surgery + RT	94
NAC + Surgery + CCRT	150
Surgery	20
Surgery + AC	249
Surgery + RT	53
Surgery + CCRT	1095
Type of surgery	
Breast-conserving surgery	608
Mastectomy	1073
Number of surveillance PET/CT scan(s)	
1	1290
2	353
3	30
4	7
7	1

NAC = neoadjuvant chemotherapy, AC = adjuvant chemotherapy, RT = radiation therapy, CCRT = concurrent chemo-radiation therapy.

**Table 2 cancers-15-02646-t002:** Number of scans according to type and location of positive finding in surveillance FDG PET/CT.

Type of Positive Scans	Local–Regional Recurrence			Distant Recurrence				2nd Primary Malignancy	
	Local, Contralateral *.	Regional Lymph Node		Bone	Thoracic †	Multiorgan	Others ‡	Thyroid	Others §
		Ipsilateral	Contralateral						
True Positive (74/105, 70.5%)	3, 1	10	4	20	17	9	4	4	2
False Positive (31/105, 29.5%)	2, 1	11	7	5	2	0	3	0	0

* Contralateral breast; † lung, mediastinal lymph nodes; ‡ liver, adrenal gland; § common bile duct cancer, lymphoma.

**Table 3 cancers-15-02646-t003:** Diagnostic performance and frequency of positive PET/CT findings according to clinical variables.

Parameter	All Scans	Interval from Last Treatment *			Prognostic Stage (AJCC 8th)			Option of NAC		Response to NAC		
		Within 2	2~4	Beyond 4	Stage I	Stage II	Stage III	NAC + Surgery	Surgery	No	Partial	Complete
	2115 Scans	729 Scans	818 Scans	574 Scans	903 Scans	486 Scans	732 Scans	345 Scans	1776 Scans	73 Scans	221 Scans	51 Scans
No. of positive scans (%)	99 (4.7%)	42 (5.8%)	34 (4.2%)	23 (4.0%)	23 (2.5%)	15 (3.1%)	61 (8.3%)	29 (8.4%)	70 (3.9%)	5 (6.8%)	19 (8.6%)	1 (2.0%)
Sensitivity (%) (No. of scans)	100 (68/68)	100 (28/28)	100 (26/26)	100 (14/14)	100 (13/13)	100 (12/12)	100 (43/43)	100 (19/19)	100 (49/49)	100 (3 /3)	100 (11/11)	100 (1/1)
Specificity (%) (No. of scans)	98.5 (2016/2047)	98.0 (685/699)	99.0 (782/790)	98.4 (549/558)	98.9 (878/888)	99.4 (469/472)	97.4 (669/687)	96.9 (314/324)	98.8 (1702/1723)	97.1 (66/68)	96.1 (198/206)	100 (50/50)
PPV (%) (No. of scans)	69 (68/99)	67 (28/42)	79 (26/33)	61 (14/23)	57 (13/23)	80 (12/15)	70 (43/61)	66 (19/29)	70 (49/70)	60 (3/5)	58 (11/19)	100 (1/1)
NPV (%) (No. of scans)	100 (2016/2016)	100 (685/685)	100 (782/782)	100 (549/549)	100 (878/878)	100 (469/469)	100 (669/669)	100 (314/314)	100 (1702/1702)	100 (66/66)	100 (198/198)	100 (50/50)
Accuracy (%) (No. of scans)	98.5 (2084/2115)	98.1 (713/727)	99.0 (808/816)	98.4 (563/572)	98.9 (891/901)	99.4 (481/484)	97.5 (712/730)	97.1 (333/343)	98.8 (1751/1772)	97.2 (69/71)	96.3 (209/217)	100 (51/51)
No. of true-positive scans	68	28	26	14	13	12	43	19	49	3	11	1
No. of false-positive scans	31	14	8	9	10	3	18	10	21	2	8	0
No. of true-negative scans	2016	685	782	549	878	469	669	314	1702	66	198	50
No. of false-negative scans	0	0	0	0	0	0	0	0	0	0	0	0

PPV = positive predictive value. NPV = negative predictive value. NAC = neoadjuvant chemotherapy. * Years.

## Data Availability

The data that support the findings of this study are available on request from the corresponding author, Y.S.C. The data are not publicly available due to the privacy of research participants.

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
