# Peer review of "Diagnostic Value of FDG PET/CT in Surveillance after Curative Resection of Breast Cancer"

_cancers, 2023, doi:10.3390/cancers15092646_

Round 1
Reviewer 1 Report
The manuscript submitted by Hwanhee Lee et al. attempted to study if the routine surveillance FDG PET/CT is useful in detection of clinically unexpected recurrence or second primary cancers with good diagnostic efficacy. To this end, the authors analysed, in a retrospective manner, 2742 consecutive female patients with curative surgically removed breast cancer and postoperative sensibilitity, specificity, PPV, NPV and accuracy of PET-CT in detecting postoperative breast cancer recurrences. The authors concluded that surveillance fluorodeoxyglucose PET/CT showed good diagnostic performance in the detection of clinically unexpected recurrent breast cancer after curative surgery.
The main strength of this study is that it addressess a relevant research question. Although current guidelines do not recommend the routine use of FDG PET/CT in follow up for breast cancer patients, the fact that these investigations have been carried out on such a massive scale in the Republic of Korea may offer a great deal of useful information on this topic, which may have significant implications for clinical practice.
However, the novelty of the subject is under question because there are also some retrospective studies on exactly the same topic. A few examples are referenced below:
Jung NY, Yoo IR, Kang BJ, Kim SH, Chae BJ, Seo YY. Clinical significance of FDG-PET/CT at the postoperative surveillance in the breast cancer patients. Breast Cancer. 2016 Jan;23(1):141-148. doi: 10.1007/s12282-014-0542-2. Epub 2014 May 29. PMID: 24872087.
Murakami R, Kumita S, Yoshida T, et al. FDG-PET/CT in the diagnosis of recurrent breast cancer. Acta Radiologica. 2012;53(1):12-16. doi:10.1258/ar.2011.110245
Title and abstract: The title and abstract are appropriate for the contents of the text. However, the authors state in line 21 that "the diagnostic performance of surveillance PET/CT was analyzed with sensitivity, specificity, positive predictive value, negative predictive value and acurracy". This sentence should be revised in order to improve the clarity of the text. Diagnostic accuracy is predominantly represented by two measures, sensitivity and specificity, so this sentence is redundant; besides, sometimes other measures, including odds ratios, likelihood ratios, are used. Alternatively, the authors might explain how accuracy is defined and measured, in order to increase the readers understanding of the text.
Also, in lines 21-22, the authors state that "findings from pathologic examination, other imaging modalities, or clinical follow-up were used as the reference standard". This sentence should be revised in order to clarify its meaning and the authors should be more specific regarding the type of reference standard imaging modalities.
Introduction: The authors summarized the current available information on the presented topic. However, several issues were encountered in this section.
In line 38, the phrase „real field clinic” should be revised.
In line 40 the authors state that „the use of laboratory or imaging examination in routine surveillance follow up has been increasing to detect asymptomatic breast cancer recurrence”. It is unclear which imaging examinations are the authors referring to? Perhaps further clarification is warranted in order to improve the readers understanding of the topic.
In line 55-57 the authors state "therefore, with large population, we performed this study to evaluate the diagnostic of routine surveillance FDG PET/CT in the detection of clinically unsuspected recurrent breast cancer after primary curative treatment" This sentence should be revised.
Materials and methods: Although the methodology for patient selection is described in detail and the inclusion and exclusion criteria are well established, several issues were encountered and warrant further clarification by the authors in order to improve the readers understanding of the study.
Results: The authors adequately presented their findings, but the way the accuracy of the proposed diagnostic imaging tool was established should be further detailed. In addition the authors fail to mention the incidence of false negative results. It is implied in the text that no false negative results were observed, which is concerning for the validity of the presented information possibly suggesting a problem of selection bias. The authors should provide more information in order to clarify this issue.
Discussions: The authors discussed their findings in relation to evidence currently available in the literature. The limitations and strengths of the present study are also appropriately discussed.
Conclusions: The conclusions of the authors should be more elaborate given the limitations of the present study.
Also at the references section between lines 282- 292 there are some annotations from the template article which should be deleted.
Lastly, while the use of language is mostly sound, a revision of grammar and syntax is required in order to improve the flow and readability of the text.
Reviewer 2 Report
Here, this study presented a retrospective large cohort study evaluating the diagnostic value of routine surveillance FDG PET/CT in breast cancer patients with the detection of clinically unexpected recurrence or second primary cancers. The work of the manuscript is of sufficient novelty and potential significance to the publication. However, several issues should be aware of before being considered accepted.
Major Issues
1. The background of this study has not been well addressed. The specific reason for launching this study has not been presented clearly in the introduction.
2. Of note, the overall frequency of positive PET/CT findings was only 5.0% and the diagnostic performance of PET/CT scans may not be significant enough to impact clinical practice in the future. Maybe the authors could make efforts to analyze the data and present better data to solidify the potential benefit of PET/CT scans in specific groups of breast cancer patients.
3. Please consider adding the results of false positive/negative findings of PET/CT in this study and discussing them properly in the discussion section.
Minor Issues
1. Molecular subtypes, histopathology, and target/endocrine therapy should also be listed in Table 1. The influence of those factors on the results should be analyzed and discussed as well.
2. In lines 68-70, the definition of surveillance PET/CT in this study is not very clear. Please explain more about the rationality of using PET/CT outside of conventional imaging. For example, if the patient has no sign or suspiciousness of recurrence or second malignancy, PET/CT scans will not be recommended in our clinical practice.
3. Normally, surveillance PET/CT was not performed concurrently with chest CT or contrast-enhanced CT due to radiation exposure and cost issues. Have those patients enrolled in this study already accepted the chest CT?
4. Is it possible to analyze the survival benefit and cost-effectiveness of surveillance PET/CT?
5. What about the time intervals between curative surgery and FDG PET/CT in this study?
6. The diversity of clinical characteristics of patients in the present study could bias the results of the study.
Round 2
Reviewer 1 Report
The authors of this paper have addressed the concerns which were previously raised. The explanation regarding the exclusion of confirmed recurrences accounts for the absence of false negative findings in this study.
Reviewer 2 Report
Please add awareness of the limitations of the conclusion in the abstract.